# Synthesis, Photophysics and Tunable Reverse Saturable Absorption of Bis-Tridentate Iridium(III) Complexes via Modification on Diimine Ligand

**DOI:** 10.3390/molecules28020566

**Published:** 2023-01-05

**Authors:** Guochang Li, Zhao Jiang, Meng Tang, Xiaoli Jiang, Houfu Tu, Senqiang Zhu, Rui Liu, Hongjun Zhu

**Affiliations:** School of Chemistry and Molecular Engineering, Nanjing Tech University, Nanjing 211816, China; 202161105047@njtech.edu.cn (G.L.); 202161105079@njtech.edu.cn (Z.J.); 201761100532@njtech.edu.cn (M.T.); 202021004032@njtech.edu.cn (X.J.); 202021004047@njtech.edu.cn (H.T.); zhuhj@njtech.edu.cn (H.Z.)

**Keywords:** bis-tridentate Ir(III) complex, photophysical property, reverse saturable absorption, optical limiting material

## Abstract

Five novel bis-tridentate Ir(III) complexes (**Ir-1**–**Ir-5**) incorporating versatile N^N^C ligands and a N^C^N ligand (1,3-di(2-pyridyl)-4,6-dimethylbenzene) were synthesized. With the combination of experimental and theoretical methods, their steady and transient state characteristics were researched scientifically. The UV-visible absorption spectra show that the broadband charge transfer absorbance of those bis-tridentate Ir(III) complexes can reach 550 nm, all of these complexes reveal the long-lasting phosphorescent emission. Because the excited-state absorption is more powerful than the ground-state absorption, a sturdy reverse saturable absorption (RSA) process can ensue in the visible and near-infrared regions when the complexes are exposed to a 532 nm laser. Therefore, the optical power limiting (OPL) effect follows the trend: **Ir-5** > **Ir-4** ≈ **Ir-3** > **Ir-2** > **Ir-1**. Generally speaking, the expansion of π-conjugation and the introduction of electron donating/withdrawing groups on the N^N^C ligand could effectively elevate the OPL effect. Therefore, these octahedral bis-tridentate Ir(III) complexes might be exploited as potential OPL materials.

## 1. Introduction

Ir(III) complexes are extensively used as organic light-emitting devices (OLEDs) [1,2,3,4], low-power up conversions [5,6], organic light-emitting electrochemical transistors [7], photocatalysts [8,9,10], and nonlinear optics [11,12,13]. This application inherently relies on their comprehensive photochemical and photophysical characteristics. The Ir(III) ion’s robust spin-orbit coupling effect leads to high-efficiency intersystem crossing (ISC) and a significant quantum yield of the triplet state [14]. Additionally, their photophysical characteristics may be tuned by structurally modifying the ligands in order to satisfy the necessities for various functions.

Organometallic Ir(III) complexes are thought to be potential nonlinear optical materials due to their extensive ground-state absorption (GSA) and strong excited-state absorption (ESA). Because the ESA is more powerful than the GSA, the reverse saturable absorption (RSA—the absorptivity of these complexes would heighten as the incoming light intensity increased) process occurs, which is one of the nonlinear absorption (NLA) behaviors [15,16,17,18]. In this respect, it has been reported that numerous Ir(III) complexes can be used as effective RSA materials [19,20]. The charge transfer (CT) driven by the D-π interaction between the metal and the chelating ligand is also of benefit to the NLA response [14]. Traditional Ir(III) complexes involving three bidentate ligands have been extensively researched in the field of optical power limiting (OPL) performance, due to their extreme synthetic versatility and tunable triplet excited properties [20,21]. Although the interest in tris-bidentate Ir(III) complexes is certainly increasing, these complexes suffer from unsatisfactory stability, causing by the poor resistance of bidentate chelate for undergoing metal–ligand dissociation [22]. Therefore, bis-tridentate Ir(III) complexes with robust architecture are thus considered to be promising OPL materials.

Since bis-tridentate Ir(III) complexes were first reported by Williams et al. [23], most of the reported works are focused on the field of OLEDs [24,25]. There is far less investigation of bis-tridentate ligands Ir(III) complexes, though interest in OPL materials is certainly increasing. Sun et al. firstly accounted for a variety of bis-cyclometalated Ir (III) complexes, and systematically investigated their NLA properties [21]. Because of the strong σ-donating ability, 2,6-diphenylpyridine (dppy) and its ramifications were selected as the bis-cyclometalating ligand(C^N^C), which endow these Ir(III) complexes with a pronounced bathochromic shift of the charge transfer (^1^CT)/^1^π,π* bands to 400−600 nm. To evaluate the feasibility of those complexes as OPL materials, our group further investigated the impact of expanded π-conjugation and replaced substituents on their RSA characteristics [20]. Realizing that the modification of tridentate chelate would certainly influence the RSA properties, the development of those Ir(III) complexes to obtain satisfactory OPL materials remains challenging.

In this work, organometallic Ir(III) complexes bearing N^N^C ligand and N^C^N ligand (1,3-di(2-pyridyl)-4,6-dimethylbenzene) were devised and synthesized ([Fig molecules-28-00566-ch001]). To avoid the bidentate N^C coordination, a pair of substituents (−CH_3_) were simultaneously inserted at the 4th and 6th locations of the central benzene ring at the same time. Phenanthroline and fluorene were employed as a building block to expand the π-conjugation of N^N^C ligand (**Ir-2** and **Ir-3**), which is due to their high structural rigidities and good charge transporting abilities. Naphthalimide and carbazole units are introduced to further explore the influence of electron donating/withdrawing groups (**Ir-4** and **Ir-5**). A suite of spectroscopic methods and theoretical models were used to investigate their physicochemical characteristics systematically. These directed Ir(III) complexes have adjustable ground-state and triplet excited-state properties, which satisfied the OPL requirements. This work gains insight into the RSA property–structure relationship among these bis-tridentate Ir(III) complexes and serves as a modified version for certain nonlinear optical absorbers [26,27,28].

## 2. Results and Discussion

### 2.1. Design and Synthesis

To adjust the ground and excited-state characteristics of the bis-tridentate complexes, different strategies were performed. To show the influences of enlarged π-conjugation, additional aromatic rings of the N^N^C ligands were inserted (**Ir-2** and **Ir-3**), while strong electron or withdrawing substituents are recommended to further satisfy the RSA properties for OPL properties. In addition, long alkyl chains were linked to both fluorene and naphthalimide groups, which were used to avoid the intermolecular aggregation and improve the solubility of the Ir(III) complexes.

The N^N^C ligands were created by combining aryl halides with phenylboronic acid in the Suzuki coupling reaction and were obtained in 50–60% yield after recrystallization. A conventional two-step technique was used to accurately produce the corresponding octahedral Ir(III) complexes **Ir-1**–**Ir-5**. This two-step method is often disseminated in mild settings, with middling yields. These complexes were all air-stable and soluble in CH_2_Cl_2_, toluene, tetrahydrofuran (THF), *N, N*-dimethylformamide (DMF), and dimethyl sulphoxide (DMSO). The predicted structure of **Ir-1**–**Ir-5** was validated by ^1^H NMR and HRMS.

### 2.2. Ground State UV-Vis Absorption

The UV-vis absorption spectra of five Ir(III) complexes in CH_3_CN are shown in Figure 1, and Table 1 summarized the detailed photophysical data. Below 300 nm, all of these complexes exhibit significant structured absorption bands, which are attributed to the intra-ligand ^1^π-π* transitions from diimine ligands. The spin-allowed charge transfer transitions, such as ^1^MLCT (metal-to-ligand changer transfer) and ^1^LLCT (ligand-to-ligand charger transfer), are responsible for the medium absorption regions at 350–500 nm. For **Ir-4** and **Ir-5**, above 500 nm, the absorption bands of the low energy tail can be referred to a mixed state, including spin-forbidden ^3^MLCT/^3^LLCT and ^3^π-π* transition [29,30]. Comparing the complexes **Ir-1**, **Ir-2,** and **Ir-3**, the increase in π-conjugation on the N^N^C ligand resulted in a red shift in the absorption band. In contrast, the electron-withdrawing/donating substituents evoked a major effect on the energy, and the absorption bands of **Ir-4** and **Ir-5** are a more bathochromic shift in comparison to those of **Ir-3**. Additionally, compared with **Ir-4**, the absorption peak of **Ir-5** is further red-shifted, and significantly increases the absorption intensity by a concomitant increase in charge transfer contribution.

### 2.3. Photoluminescence

To further comprehend the excited-state features, the emission characteristics of these Ir(III) complexes were also investigated. Figure 2 depicts the normalized emission spectra of **Ir-1**–**Ir-5**, and the corresponding parameters are reported in Table 1. All of the complexes have congruent room temperature phosphorescence, which is responsive to oxygen, and also have relatively long lifetimes. Their excitation spectra are shown in Appendix A. It is obvious that their excited-state characteristics are influenced by the modified N^N^C ligands. Although these complexes own the wide and structureless emission bands, their emission lifetimes are different. For **Ir-1**–**Ir-3,** the expansion of π-conjugation has little influence (111 ns for **Ir-1**, 105 ns for **Ir-2**, and 135 ns for **Ir-1**). However, comparing **Ir-3**–**Ir-5**, the introduction of electron-withdrawing/donating substituents prolong their lifetimes (241 ns for **Ir-4** and 341 ns for **Ir-5**), which is due to the intramolecular charge transfer. The emission properties of solvents with different polarities are further studied (Figure 2b–f). Taking **Ir-1** for example (Figure 2b), it was discovered that the low energy emission bathochromically moved to longer wavelengths in solvents with higher polarity (e.g., CH_3_CN) relative to its location in the less polar solvents (e.g., THF). Considering their wide and structureless emission bands and solvatochromic effect (As show in Table 2), the emission properties of these complexes can be mainly ascribed to ^3^MLCT/^3^LLCT states with little ^3^π-π* state [31,32,33].

### 2.4. DFT Calculations

To gain a better understanding of the structure–property relationships of these complexes, the DFT calculation was carried out to acquire a thorough grasp of the underlying foundation of the aforementioned experimental findings [34,35]. Table 3 and Table 4 list the excitation energies, wavelengths, oscillator strengths, associated configuration coefficients, and the major principal configuration of the lowest electronic states (highest occupied molecular orbitals (HOMOs) and lowest unoccupied molecular orbitals (LUMOs)). From Table 3, the optical transitions of S_1_ and T_1_ states of complexes **Ir-1**–**Ir-5** are primarily from HOMO-1→LUMO and HOMO→LUMO transitions. As shown in Table 4 and Appendix A, the electron density of the HOMO of the complexes **Ir-1** and **Ir-2** is mainly distributed on N^C^N and the center Ir atom, and the LUMO is almost exclusively located in N^N^C ligand (2,2′-bipyridine of **Ir-1** and 1,10-phenanthroline of **Ir-2**). However, the electron density of the HOMO of the complex **Ir-3** is mainly distributed on the N^N^C ligand fluorene and Ir atom, while the LUMO is distributed the N^N^C ligands and the central Ir atom. In addition, the LUMO distributions of **Ir-4** and **Ir-5** share similar properties, while the electron density of the HOMO of the complexes is mainly distributed on the N^N^C ligand, including fluorene and terminal group (naphthalimide on **Ir-4** and carbazole on **Ir-5**). Although the orbital configurations are complicated, the major transitions of all target complexes are associated with the MLCT and LLCT transitions. Moreover, the trends of the calculated S0→S1 transition wavelengths and T1→S0 transition wavelengths for these complexes are similar to the commencement of UV-Vis absorption onset and photoluminescence.

### 2.5. Transient Absorption

Nanosecond transient (TA) absorption of **Ir-1**–**Ir-5** were researched to further comprehend their triplet excited-state properties in order to determine the spectral region where RSA may happen. The nanosecond time-resolved TA spectra for complexes **Ir-1**–**Ir-5** are presented in Figure 3. Table 1 lists the lifetimes of the triplet excited states inferred from the TA decay profile and quantum yields.

These complexes exhibit broad and strong positive absorption bands ranging from ~440 to 800 nm, as seen in Figure 3, which means that the RSA could occur in these ranges. It is noted that the TA spectra of **Ir-1** and **Ir-2** show the sharp and intense TA bands at ca. 550 nm with short lifetimes, which is due to ^3^π−π* state from diimine ligands. The lifetimes of the triplet excited states determined by the decay of the transient absorption spectra are 69~86 ns, which are congruous with their emission lifetimes. These characters indicate that the origins of the triplet excited absorption states are the same as those of the emission states. Although the complex **Ir-3** shares a similar lifetime, the sharp and intense TA bands disappear because of the increase in charge transfer contribution. On the other hand, the connected electron-donating/withdrawing groups associated with the N^N^C ligands have a significant impact on the intensity and maximum absorption peak of TA for these Ir(III) complexes. Considering that the electron density of the HOMO of the complexes is mainly distributed on the fluorene and terminal groups, it can be conjectured that the observed TA signals from complexes **Ir-4** and **Ir-5** are thought to be mostly caused by the ^3^MLCT/^3^LLCT T1 states. Additionally, the introduction of terminal groups efficaciously prolongs the triplet excited lifetimes (179 ns at 530 nm for **Ir-4** and 341 ns at 530 nm for **Ir-5**). It should be pointed out that the complex **Ir-5** presents the longest triplet excited lifetimes and strongest TA signals intensity, especially after 600 nm. As demonstrated in Figure 3f, the broad TA band of **Ir-5** kept increasing in the NIR regions, which expands the working window of RSA.

### 2.6. Optical Power Limiting Properties

All complexes show stronger ESA than the GSA, and the broad positive TA absorption bands (Figure 3) reveal that the RSA process would occur in these ranges. In order to assess the OPL response, the target Ir(III) complexes were irradiated with a 532 nm Nd:YAG laser [13,15,36,37]. For comparison purposes, the concentration in solution was set to 80% transmittance at 532 nm for all samples. Figure 4 demonstrated that all complexes exhibit better OPL performance than the standard OPL material (C_60_). The output energy density reduced dramatically as the incoming laser’s energy increased. As shown in Figure 4 and Table 5, the intensity of the OPL response corresponds to the trend **Ir-5** > **Ir-4** ≈ **Ir-3** > **Ir-2** > **Ir-1**. The complex **Ir-5** exhibited the best OPL ability; when the energy density of the input was raised to 1.5 cm^−2^, the transmittance is only 20%. **Ir-5** has potential to be the broadband reverse saturable absorber in the visible–near-infrared spectral range. Compared with other bis-tridentate Ir(III) complexes, modifying electron-donating units or extending π-conjugation of the ligands is a systematic way to improve the RSA effect [20,21,38].

Based on the analysis and discussion above, Figure 1 summarizes the RSA processes involved in the Ir(III) complexes. Under 532 nm laser excitation, the Ir(III) complexes, **Ir-1**–**Ir-5**, were excited to the S_1_ state by 520 nm laser radiation. Then, Ir(III) complexes were stimulated to a higher singlet state (S_n_) or under heavy-atom-effect-induced ISC to the first triplet state (T_1_). The Ir(III) complexes on the T_1_ state could further absorb the photons to a higher triplet state (T_n_) under the laser radiation. When the ESA is stronger than GSA, there is an effective NLA response based on the RSA process.

## 3. Experimental Section

### 3.1. Computational Methods

The ground electronic state frontier molecular orbitals (FMOs) of complexes **Ir-1**–**Ir-5** were simulated by density functional theory (DFT), and their excited electronic states were calculated using the time-dependent density function theory (TDDFT) method. All calculations were performed using the Gaussian 09 software package. The specific DFT method used was PBE1PBE for calculating excited electronic states. The basic sets used in all calculations were the 6–31G* set for all light atoms and the LANL2DZ set for Ir(III) atoms. Full equilibrium geometry optimization was performed for the ground electronic states of all complexes. For consistency with experiments, acetonitrile (CH_3_CN) used for experimental measurements was selected as the solvent in theoretical calculations as well.

The materials and measurements are described in depth in the Appendix A. Figure 2 outlines the synthesis of the Ir(III) complexes. **L0** was synthesized according to methods in the literature [20]. Ligands **L1**–**L5** are also presented in Appendix A.

### 3.2. Preparation of Complexes

Preparation of the Ir chloride complexes. IrCl_3_ (200 mg, 0.6 mmol), **L0** (0.6 mmol), ethylene glycol (30 mL), and water (10 mL) were added to a 100 mL round-bottom flask under nitrogen. The mixture was refluxed (120 °C) and stirred for 24 h. Then, using water (30 mL) to quench the reaction, the suspension was filtered. The crude product was dried at 45 °C under vacuum. Yellow solid was obtained (183 mg, yield 45%). Without further confirmation, Ir(III) chloride complexes were used in the next reaction.

Preparation of the complexes **Ir-1**–**Ir-5**. The µ-chloro-bridged dimer intermediate (523 mg, 0.5 mmol), **L1**–**L5** (1 mmol), ethylene glycol (30 mL), and AgCF_3_SO_3_ (617 mg, 2.4 mmol) were added to a 100 mL round-bottom flask under nitrogen. The mixture was refluxed with stirring for 24 h. Then, after cooling to room temperature, excess KPF_6_ was added and stirred at room temperature for 3 h. The residue was taken into H_2_O and extracted with dichloromethane three times. The remnant was dried over anhydrous Na_2_SO_4_ and concentrated. The mixture was purified by flash chromatography on a silica gel using dichloromethane and methanol (100:1, *v*/*v*) as the eluent.

Complex **Ir-1**. Yellow solid (203 mg, yield 33%). ^1^H NMR (400 MHz, DMSO-*d_6_*) δ 8.86–8.75 (m, 2H), 8.59 (d, *J* = 8.1 Hz, 1H), 8.39 (t, *J* = 8.1 Hz, 1H), 8.23 (d, *J* = 8.4 Hz, 2H), 8.10 (td, *J* = 8.0, 1.4 Hz, 1H), 7.91 (d, *J* = 7.3 Hz, 1H), 7.88–7.79 (m, 2H), 7.49 (dd, *J* = 10.1, 5.0 Hz, 3H), 7.40–7.33 (m, 1H), 7.18 (s, 1H), 6.98 (t, *J* = 6.5 Hz, 2H), 6.83 (t, *J* = 7.1 Hz, 1H), 6.62 (t, *J* = 7.4 Hz, 1H), 5.78 (d, *J* = 7.1 Hz, 1H), 2.92 (s, 6H). HRMS (ESI) m/z calcd for C_34_H_26_IrN_4_^+^ (M)^+^ 683.1781, found 683.1785.

Complex **Ir-2**. Yellow solid (203 mg, yield 32%). ^1^H NMR (400 MHz, DMSO-*d_6_*) δ 8.90 (q, *J* = 8.8 Hz, 2H), 8.66 (dd, *J* = 8.3, 1.2 Hz, 1H), 8.44 (d, *J* = 8.9 Hz, 1H), 8.23 (d, *J* = 8.9 Hz, 3H), 8.04 (t, *J* = 7.5 Hz, 1H), 7.85 (dd, *J* = 5.0, 1.2 Hz, 1H), 7.82–7.75 (m, 2H), 7.61 (dd, *J* = 8.2, 5.0 Hz, 1H), 7.40–7.33 (m, 2H), 7.20 (s, 1H), 6.92–6.86 (m, 1H), 6.86–6.78 (m, 2H), 6.65 (td, *J* = 7.4, 1.3 Hz, 1H), 5.92–5.84 (m, 1H), 2.93 (s, 6H). HRMS (ESI) m/z calcd for C_36_H_26_IrN_4_^+^ (M)^+^ 707.1781, found 707.1787.

Complex **Ir-3**. Yellow solid (203 mg, yield 32%). ^1^H NMR (400 MHz, DMSO-*d_6_*) δ 8.75 (ddd, *J* = 11.1, 10.4, 5.5 Hz, 3H), 8.38 (dd, *J* = 9.5, 6.6 Hz, 1H), 8.23 (t, *J* = 9.8 Hz, 2H), 8.16 (t, *J* = 7.1 Hz, 1H), 8.08 (t, *J* = 7.6 Hz, 1H), 7.85–7.72 (m, 2H), 7.46 (dt, *J* = 10.7, 5.0 Hz, 3H), 7.38–7.33 (m, 1H), 7.31 (d, *J* = 6.9 Hz, 1H), 7.24 (s, 1H), 7.15 (d, *J* = 9.2 Hz, 1H), 7.09 (dd, *J* = 11.8, 7.2 Hz, 2H), 6.93 (t, *J* = 5.7 Hz, 2H), 6.11 (t, *J* = 2.6 Hz, 1H), 2.96 (d, *J* = 3.0 Hz, 6H), 2.09–1.78 (m, 4H), 0.82–0.11 (m, 30H). HRMS (ESI) m/z calcd for C_57_H_62_IrN_4_^+^ (M)^+^ 995.4598, found 995.4597.

Complex **Ir-4**. Yellow solid (153 mg, yield 30%). ^1^H NMR (400 MHz, DMSO-*d_6_*) δ 8.78 (ddd, *J* = 10.4, 8.1, 2.7 Hz, 3H), 8.55–8.47 (m, 2H), 8.41 (t, *J* = 7.9 Hz, 1H), 8.26 (dd, *J* = 9.7, 6.3 Hz, 3H), 8.14–8.07 (m, 1H), 8.07–8.00 (m, 1H), 7.87–7.77 (m, 2H), 7.71 (t, *J* = 7.8 Hz, 1H), 7.63 (dd, *J* = 7.5, 1.5 Hz, 1H), 7.53–7.44 (m, 4H), 7.42–7.35 (m, 1H), 7.29 (s, 2H), 7.25 (s, 1H), 6.99–6.90 (m, 2H), 6.21 (d, *J* = 3.9 Hz, 1H), 4.05 (t, *J* = 7.2 Hz, 2H), 2.97 (s, 6H), 2.12–1.88 (m, 4H), 1.69–1.55 (m, 2H), 1.40–1.28 (m, 2H), 0.92 (t, *J* = 7.3 Hz, 3H), 0.77–0.18 (m, 30H). HRMS (ESI) m/z calcd for C_73_H_75_IrN_5_O_2_^+^ (M)^+^ 1246.5544, found 1246.5535.

Complex **Ir-5**. Yellow solid (203 mg, yield 32%). ^1^H NMR (400 MHz, DMSO-*d_6_*) δ 8.84–8.72 (m, 3H), 8.58–8.49 (m, 2H), 8.41 (t, *J* = 7.8 Hz, 1H), 8.27 (dd, *J* = 13.7, 5.9 Hz, 3H), 8.14–8.03 (m, 2H), 7.83 (dd, *J* = 17.8, 8.9 Hz, 2H), 7.73 (t, *J* = 7.8 Hz, 1H), 7.65 (d, *J* = 7.5 Hz, 1H), 7.57–7.43 (m, 4H), 7.39 (d, *J* = 5.6 Hz, 1H), 7.31 (s, 1H), 7.26 (s, 1H), 6.96 (dd, *J* = 12.0, 5.9 Hz, 2H), 6.21 (d, *J* = 3.1 Hz, 1H), 4.07 (t, *J* = 7.1 Hz, 2H), 2.98 (s, 6H), 2.51 (s, 18H), 2.10–1.91 (m, 4H), 0.87–0.22 (m, 30H). HRMS (ESI) m/z calcd for C_77_H_85_IrN_5_^+^ (M)^+^ 1272.6428, found 1272.6435.

## 4. Conclusions

Five bis-tridentate Ir(III) complexes with different π-conjugation and terminal groups on the N^N^C ligands have been synthesized. In order to better explore the RSA properties, their photophysical characteristics have been thoroughly studied utilizing spectroscopic methods. All Ir(III) complexes exbibit intra-ligand ^1^π-π* transitions and ^1^LLCT/^1^MLCT transition. The modification on the N^N^C ligands generated a bathochromic shift in the absorption band because of the increase in charge transfer contribution. The emission properties of these complexes are from the triplet excited state and can be mainly attributed to ^3^MLCT/^3^LLCT states with little ^3^π-π* state. In addition, the introduction of terminal groups efficaciously prolongs the triplet excited lifetimes. Broadband triplet TA is observed in these complexes in the visible-NIR region. Considering their significantly broader GSA and higher ESA, all complexes produce a substantial NLA response, which follows the descending order **Ir-5** > **Ir-4** ≈ **Ir-3** > **Ir-2** > **Ir-1**. Therefore, these Ir(III) complexes are promising nonlinear absorber materials for the application of OPL.

## Data Availability

Not applicable.

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
