# Peer review of "Synthesis, Photophysics and Tunable Reverse Saturable Absorption of Bis-Tridentate Iridium(III) Complexes via Modification on Diimine Ligand"

_molecules, 2023, doi:10.3390/molecules28020566_

Round 1
Reviewer 1 Report
Guochang Li et al submitted an article having the title “Synthesis, Photophysics and Tunable Reverse Saturable Absorption of Bis-tridentate Iridium(III) Complexes via Modification on Diimine Ligand” to on Molecules. This article explains to discuss synthesis and photophysical properties of iridium complexes. Thus I recommended the article for the publication with major revision. The following points need to be noted and revised prior to publication,
1 1. Authors should include both experimental and computational parameters in Table 3, also discuss correlation between both the parameters.
2. How did authors predict excited state computationally?
3. How much electron density is distributed on the fluorene and terminal groups in all complexes? This detailed analysis will provide more insight for the readers to understand their application
4. I could not able to supporting information, so not able to see the methodology part. Since this is full paper, so author should include computational methodology in the main manuscript.
5. There is no concrete conclusion why Ir-5 > Ir-4 ≈ Ir-3 > Ir-2 > Ir-1.
6. Authors should cite more recent references.
Author Response
Dear Reviewer,
Enclosed please find our revised manuscript “Synthesis, Photophysics and Tunable Reverse Saturable Absorption of Bis-tridentate Iridium(III) Complexes via Modification on Diimine Ligand” (Manuscript ID: molecules-2091065). We appreciate the opportunity that you provided for us to revise our manuscript, and the valuable suggestions. Our responses to the reviewer’s comments are detailed below. All the revisions are highlighted in red in the revised manuscript.
Point 1: Authors should include both experimental and computational parameters in Table 3, also discuss correlation between both the parameters.
Response: Thanks for your nice comment and suggestion, we added the both experimental and computational parameters in Table 3 for comparison, which have been discussed in the manuscript. The experimental results are in good agreement with the calculated results. The corresponding comment and discussion have been revised in the manuscript and highlighted in red.
Point 2: How did authors predict excited state computationally?
Response: Sorry for the confusion, all calculations were performed using the Gaussian 09 software package. In order to understand the contribution of the complexes in the excited state, the time-dependent density function theory (TDDFT) calculations were used to calculate these complexes. The specific TDDFT method used was PBE1PBE for calculating excited electronic states. The basic sets used in all calculations were the 6-31G* set for all light atoms and the LANL2DZ set for Ir(III) atoms. These computational methods were provided in the Experimental section.
Point 3: How much electron density is distributed on the fluorene and terminal groups in all complexes? This detailed analysis will provide more insight for the readers to understand their application.
Response: Thanks for your nice suggestion. The contribution of electron density for these complexes was provided in Table S1 (ESI). Detailed analysis and discussion were added in Section 3.4.
Point 4: I could not able to supporting information, so not able to see the methodology part. Since this is full paper, so author should include computational methodology in the main manuscript.
Response: The ESI file has been uploaded in the submission system again. According to your suggestion, the computational methodology has also been added in the Experimental section of the main manuscript.
Point 5: There is no concrete conclusion why Ir-5 > Ir-4 ≈ Ir-3 > Ir-2 > Ir-1.
Response: Sorry for the confusion. The optical limiting threshold was defined as the incident energy density at a transmittance of 50% (F50). As shown in Figure 4 and Table 5, the intensity of the OPL response corresponds to the trend of Ir-5 > Ir-4 ≈ Ir-3 > Ir-2 > Ir-1.
Point 6: Authors should cite more recent references.
Response: Thanks for your suggestion, valuable and rfecent references have been cited in the manuscript.

Reviewer 2 Report
1. “Above 500 nm, the absorption bands of the low energy tail can be referred toa mixed state including spin-forbidden 3MLCT/3LLCT and 3π-π* transition.” How did you support such results?
2. What’s the means of MLCT/ LLCT?
3. Please give the exciting emission for the five complex
4. To gain a better understanding the structure-property relationships of these complexes, the DFT calculation was carried out to acquire a thorough grasp of the underlying foundation of the aforementioned experimental findings. Please add the ref on the DFT, also, this part can be referred to Coord. Chem. Rev., 445(2021) 214074; Coord. Chem. Rev. 2020, 406:213145 AND CrystEngComm, 2021, 23, 8043–8052
5. The optical power limiting (OPL) effect follows the trend: Ir-5 > Ir-4 ≈ Ir-3 > Ir-2 > Ir-1.The authors should compare similar results that have been reported, and cite relative conclusion to support such data.
Round 2
Reviewer 1 Report
The revision produced by authors is improved manuscript and most of the questions raised by the reviewer has been addressed satisfactorily and lead to an manuscript that is acceptable for publication.